# Recent Advances in Nanomicelles Delivery Systems

**DOI:** 10.3390/nano11010070

**Published:** 2020-12-30

**Authors:** Salah M. Tawfik, Shavkatjon Azizov, Mohamed R. Elmasry, Mirkomil Sharipov, Yong-Ill Lee

**Affiliations:** 1Department of Materials Convergence and System Engineering, Changwon National University, Changwon 51140, Korea; salahtawfik85@gmail.com (S.M.T.); shavkat9191@gmail.com (S.A.); m.elmasry.chem2014@gmail.com (M.R.E.); mirkosharipov@gmail.com (M.S.); 2Surfactant Laboratory, Department of Petrochemicals, Egyptian Petroleum Research Institute (EPRI), Nasr City, Cairo 11727, Egypt; 3Laboratory of Polysaccharide Chemistry, Institute of Bioorganic Chemistry, Uzbekistan Academy of Science, Tashkent 100125, Uzbekistan

**Keywords:** nanomicelles, spectroscopy, drug delivery, fluorescence imaging, stimulus-responsive

## Abstract

The efficient and selective delivery of therapeutic drugs to the target site remains the main obstacle in the development of new drugs and therapeutic interventions. Up until today, nanomicelles have shown their prospective as nanocarriers for drug delivery owing to their small size, good biocompatibility, and capacity to effectively entrap lipophilic drugs in their core. Nanomicelles are formed via self-assembly in aqueous media of amphiphilic molecules into well-organized supramolecular structures. Molecular weights and structure of the core and corona forming blocks are important properties that will determine the size of nanomicelles and their shape. Selective delivery is achieved via novel design of various stimuli-responsive nanomicelles that release drugs based on endogenous or exogenous stimulations such as pH, temperature, ultrasound, light, redox potential, and others. This review summarizes the emerging micellar nanocarriers developed with various designs, their outstanding properties, and underlying principles that grant targeted and continuous drug delivery. Finally, future perspectives, and challenges for nanomicelles are discussed based on the current achievements and remaining issues.

## 1. Introduction

Progress in the development of new nanotechnology-based approaches and efficient drug-delivery carriers has enabled targeting therapeutic molecules to target tissues. To achieve an efficient therapeutic response, two important factors should be taken into consideration. The first factor is the delivery of the necessary amount of active compound to the target site and the second factor is to maintain the effective dose for a certain time. However, it is difficult to control these factors for most drugs due to various insuperable barriers, including imperfect pharmacokinetics, loss of specificity for a target, and extensive biodistribution. Up to date, most developed anticancer drugs are hydrophobic and their biodistribution in body limits the dose because of the side effects. Therefore, the targeted nanomedicines based on controlling the drugs delivery and reducing toxicity have become a popular field of research. This research aims to develop and design effective approaches to overcome existing medical issues and to establish the basis for unprecedented or revolutionary treatments [1,2,3].

Up until a few years ago, a different class of nanomedicine targeting strategies based on liposomes, dendrimer, and micelles is promoted as promising drug delivery systems (DDSs) for controlled, sustained, and targeted delivery due to their properties to reduce the toxicity toward nontargeted tissues and lead to the accumulation of drugs at the targeted site [4,5,6,7]. Among these DDSs, nanomicelles have received tremendous interest in nanomedicine due to their low-cost, great biocompatibility, simple preparation methods, and effectiveness [8]. Moreover, nanomicelles compared to conventional micelles are more thermodynamically stable in physiological solutions. This stability is indicated by the lower critical micellar concentration (CMC), which is essential for the stability and slow dissociation in vivo microenvironment [9]. As reported, the nanomicelles have shown lower dissociation kinetics after dilution around 1000-fold slower compared to conventional surfactant micelles. This property is essential for DDSs, which circulate in the bloodstream until their accumulation in the target site [10,11].

Significant efforts have been devoted to ensuring the accumulation of DDSs in the target location via the conjugation of nanomicelles with specific ligands such as aptamers, antibodies, and other biological molecules [12,13,14,15]. Any disease and disorder is followed by changes in the microenvironment of the unhealthy tissues that can be used in targeting strategies. One of the popular targeting materials is folic acid (FA), which binds to cancer cells, which overexpress folic receptors. For instance, Park and coworkers have encapsulated doxorubicin (DOX) inside the (PEG–poly[lactide-co-glycolide]) nanomicelles conjugated with folate to ensure selective delivery [16]. In a similar study, nonionic alginate-based polymers and FA have been developed by Tawfik et al. and showed that when the particle is covered by FA, it can be easily entrapped into KB cells by the folate receptors [17].

To expand the performances of DDSs, various endogenous and exogenous stimulations-responsive functionalities have been introduced. Characteristics of microenvironment, such as pH, expression of enzyme, and temperature can vary in normal and disease state, thus they can be employed as endogenous stimulation. Another category of stimulation is exogenous stimulation, including redox potential, magnetism, light, and ultrasound triggering. This method can promote the release of the drug at a specific location via specific stimulation. Moreover, this strategy provides improved drug release, enhances therapy, controls some activity on the microenvironment of the target area [18,19,20]. Additionally, imaging and delivery systems utilizing photoluminescence, particularly fluorescent nanomicelles are becoming desirable as possible cancer screening tools [17,21,22].

In this review, we summarize recently developed micellar nanocarriers with various designs, their outstanding properties, and underlying strategies that grant targeted and continuous drug delivery system and discuss the discoveries and achievements in the field of nanomicelles that have shown their potential, specifically in the delivery of anticancer drugs for cancer therapy.

## 2. Nanomicelles

Amphiphilic molecules are common subunits of supramolecular assembly such as liposomes, micelles, and nanomicelles due to the possession of both hydrophilic (polar) and hydrophobic (nonpolar) groups. The assembly of these molecules occurs due to the orientation of the groups to a suitable environment such as a solvent. For instance, if the solvent is polar, the hydrophilic portion of molecules orients toward the outer surface to maximize contact with a polar solvent, while the hydrophobic parts are clustered in the core to minimize contact with a polar solvent [21,23,24]. This kind of supramolecular assembly is referred to as normal nanomicelles. Alternatively, the supramolecular assembly formed in the nonpolar solvent is known as reverse nanomicelles, in which the hydrophobic portion orients to the surface to maximize the contact with the solvent and hydrophilic portion orients to the core to dodge the contact with a solvent. The property to change the orientation depending on the solvent allows loading different types of the drug into nanomicelles. Nonsoluble drugs can be loaded into normal nanomicelles while soluble drugs can be loaded in reverse nanomicelles [25]. Nanomicelles act as a protective shell by reducing the direct contact of drugs with the in vivo environment, thus they improve drug bioavailability and reduce adverse side effects [26,27,28].

## 3. Surfactant Nanomicelles

Surfactant is one category of amphiphilic molecules that have hydrophilic heads and hydrophobic tails. Surfactant tends to form a supramolecular assembly known also as a colloidal dispersion, which has a small diameter in the range from 5 to 100 nm [29]. The control of the size of the micelles is based on the characteristics of the head group and the variation of the alkyl chain length [30,31]. Head groups are classified in different groups such as charged (anionic or cationic), dipolar (zwitterionic), and noncharged (nonionic) [32]. In aqueous solution, surfactant orients with head to the solvent and first accumulates on the air-liquid interfaces. By increasing the amount of surfactant, we induce the self-assembly of surfactants into micelles (Figure 1). Moreover, the hydrophobic core is formed via the van der Waals bond recognition [29]. The concentration above which the surfactant starts to self-assemble into micelles is known as critical micellar concentration (CMC). The formed micelles will set up a hydrogen bond crosslink between the head group and water molecules from aqueous solution.

Self-assembled micelles can have various structures such as a sphere, cylinders, lamellar, and vesicles based on the characteristics of the medium, length of chain, and temperature (Figure 2) [9,34,35]. Low molecular weight surfactants have considerably higher CMC, which is 10^−3^ to 10^−4^ M, compared to polymeric amphiphilic molecules, which has a CMC located in the range of 10^−6^ to 10^−7^ M. Micelles with lower CMC are preferable in DDSs because of their relatively higher stability and insensitivity to the dilution, which enables their longer circulations in the bloodstream [36]. Functionalizing the polymeric micelles with polyethylene oxide (PEO) or polyethylene glycols (PEG) group results in physically blocking the interparticle interaction between core regions and hinders interactions between the core-forming blocks and blood components. Especially, the long chain of PEO and high surface density lead to minimizing protein adsorption to hydrophobic surfaces [37]. For polymeric micelles, the length of the PEO chains affects circulation time and uptake by the reticuloendothelial system, with longer chains prolonging circulation time and minimizing reticuloendothelial system uptake [38]. Because PEO blocks are relatively inert and provide “stealth” characteristics to polymeric micelles, encapsulation in polymeric micelles may be a promising technology to prolong the circulation time of therapeutic agents [39]. Similarly, the key role of PEGylated-based micelles was described by Zhang et al. who successfully prepared a novel redox-responsive-nanoparticles (MPSSV-NPs) via self-assembled of PPM-vitamin E conjugate and a PEG derivative of linoleate (mPEG2000-LA) in aqueous solution. The pharmacokinetics of PSSV-NPs and MPSSV-NPs have displayed a prolonged circulation time with an elimination phase half-life (t_1/2_) of 3.339 and 30.04 h, respectively, compared to PPM with 0.327 h [40].

## 4. Polymeric Nanomicelles

Another category is polymeric nanomicelles (PNMs) prepared from amphiphilic polymers; they have a size that ranges from 1 to 200 nm. This type of nanoparticles contains two functional portions, an “inner core” and an “outer shell”. The outer shell is responsible for controlling the pharmacokinetic properties in vivo, thus, it consists of a hydrophilic block of polyethylene glycol (PEG). Further modification of the outer shell can improve the properties of nanomicelles, for example, enhance the targeting. The inner core consists of a hydrophobic core that is responsible for drug entrapment, the stability of nanomicelles, and drug-release characteristics [42,43]. As reported, polymeric micelles with high stability have shown good biocompatibility in cancer therapy applications. Moreover, PNMs can encapsulate poorly soluble pharmaceuticals and enhance their biological behavior [44,45]. Especially, anticancer drugs encapsulated in polymeric micelles have shown a better accumulation in the targeted site, thus reducing side effects on noncancerous tissues [46,47,48]. Moreover, drugs entrapped in micelles are protected from unexpected interactions and biodegradation in vivo environment, resulting in prolongation of the half-life and improved efficiency [49,50,51]. The surface of nanoparticles has two important functions, protection of drugs from interaction with the in vivo environment and delivering it to the target site. Some commonly used micellar carriers are shown in Figure 3.

### 4.1. Polymer Selection, Synthesis, and Properties

#### 4.1.1. Hydrophilic Polymers in Block Copolymers

The modification of hydrophilic shells has been widely studied and physicochemical characteristics of hydrophilic micelles such as surface density and molecular weight have been closely linked to stability, circulation time, and good biodistribution of micelles in vivo [53].

##### Poly(ethylene glycol)

One of the widely used hydrophilic polymers is PEG (Mw ~1–15 kDa). This polymer has acquired popularity in this field due to the important characteristics such as physicochemical, good water solubility, chain mobility, and great biocompatibility. Several alternative polymers have also shown good characteristics. For instance, poly(*N*-vinyl-2-pyrrolidone) (PVP) [54] has been used in liposomes [55] and diblock polymer micelles [56] an alternative to PEG due to its high biocompatibility.

Low molecular weight PEG (20 kDa) have low toxicity [57]. PEG has been the gold standard for nanomedicine polymers. PEG with molecular weights (1 to 6 kDa) is an excellent molecular weight for featuring nanoparticles with powerful antifouling characteristics and is commonly used to synthesize block copolymers for drug delivery [58]. Both the molecular weights and surface density of PEG are important factors when creating the shell. It was reported that PEG conformation significantly influences the circulation time and clearance of polymeric micelles in vivo [53]. PEG synthesis is commonly done by anionic ring opening polymerization (ROP) of ethylene oxide and this synthetic method produces a well-defined PEG with a narrow size distribution [58]. The structural functionality of PEG can be enhanced by the modification of the end group of PEG by suitable chemical reagents. The chemical diversity of the PEG end group includes additional reactive moieties for the labeling of ligands that allow further conjugation with other species of micelles to synthesize target block copolymers.

##### Hydrophilic Poly(2-oxazoline)s

Due to their biocompatibility and chemical flexibility, poly(2-oxazoline) (POx)-based block copolymers have recently gained a lot of attention as novel biomaterials [59]. The antifouling characteristics of hydrophilic POx such as poly(2-methyl-2-oxazoline) (PMeOx) and poly(2-ethyl-2-oxazoline) (PEtOx) have been shown to prevent rapid clearance by in vivo RES. The ability of this hydrophilic POx was demonstrated in these studies [60]. Via living cationic ring opening polymerization (LCRP), POx can be easily synthesized and latest block copolymers composed of POx have shown scalable synthesis and chemical versatility [59]. The PMeOx and PEtOx displayed extremely low protein adsorption and cell adhesion that is comparable to that of PEG-coating as reported by Zhang et al. [61]. Remarkably, the functionalization on the end group of those polymers showed a small effect on the protein adsorption properties, unlike with PEG. Nevertheless, the length and molecular weight of the polymer was considerably related to the antifouling properties. POx hydrophilic polymers have demonstrated enhanced mucus penetrating properties which could be important in the oral delivery of polymer micelles [62]. They exhibited that PMeOx had superior muco-penetrating properties, as determined by the diffusion coefficient in gastric mucus, compared to silica nanoparticles. PEtOx also displayed some muco-penetrating improvement but less so than PMeOx.

##### Other Reported Hydrophilic Polymers

Hydrophilic poly(amino acid)s have been used as amphiphilic block copolymers in micelles to create the outer shell of the polymeric micelles. The biodegradability of poly(amino acid)s in vivo potentially confers the safety of these micelles in the body [63]. However, this could mean that antifouling properties are not maintained for long times like those observed with POx systems. Preparation of hydrophilic poly(amino acid)s can be performed via anionic ROP utilizing the *N*-carboxyanhydride of amino acids to produce poly(aspartic acid) (P(Asp)), poly(glutamic acid) (P(Glu)), and poly(sarcosine) [64]. Among the hydrophilic poly(amino acid)s, poly(sarcosine) has presented good antifouling properties in recent reports [65].

Chitosan, Dextran, heparin, hyaluronic acid, and chondroitin sulfate polysaccharides have also displayed good properties and inhibited protein adsorption on the surface of the particles in biological fluids. Importantly, some reports indicated dextran as a shielding agent for nanoparticles and prolonged circulation in vivo [66]. Several reports have studied the micellar properties of PVP. PVP can be prepared via radical polymerization, and it has commonly been utilized as a hydrophilic polymer in micelles formulation design. Apparently, both the pyrrolidone moiety and amide groups in the side chain are closely related to the antifouling properties of PVP, but comprehensive mechanisms of these properties are still unknown [67]. Several other zwitterionic polymers have been developed as antifouling macromolecules [68]. Those polymers are expected to be useful for the development as block copolymers for the efficient delivery of poorly soluble drugs in polymeric micelles formulations.

#### 4.1.2. Hydrophobic Polymers in Block Copolymers

Generally, the hydrophobic core consists of polyesters, polyethers, and polyamino acids. The major benefit of polyester over other polymers is that a wide range of polyesters was approved for biomedical usage, such as poly(lactic acid) (PLA), poly(ε-caprolactone) (PCL), poly(propylene oxide) (PPO), poly(trimethylene carbonate) (PTMC), and poly(lactic-co-glycolic acid) (PLGA) [69]. These biodegradable hydrophobic polyester blocks are responsible for the formation of the hydrophobic core and they are usually conjugated with the hydrophilic block (mainly PEG), which forms an outer shell of micelles. Other well-known hydrophobic blocks that have a promising bioavailability are polyanhydrides blocks. Poly sebacic anhydride (PSA) belongs to the family of polyanhydrides. These hydrophobic blocks also have a PEG chain and showed low toxicity and further biodegradation into nontoxic products [70]. The poly(l-amino acid) (PAA) is a large class of polymers that consists of l-amino acids, including poly(l-aspartic acid) (polyAsp), poly(l-lysine)(polyLis), and poly(l-glutamic acid) (polyGlut). These hydrophobic polymers have been reported to form nanovesicles after covalent conjugation with PEG. Moreover, they have demonstrated not only efficient biodegradability and biocompatibility [70] but also have pH sensitivity [71]. Tumors have usually acidic microenvironment due to higher proliferation activity that is beneficial for pH-responsive drug delivery system. Phospholipid-based blocks conjugated with PEG, including PEG-distearoylphos-phatidylethanolamine (DSPE), have also been reported to self-assemble in nanostructured micelles. Advantages of this family-based micelles are easy to prepare, have good biocompatibility, and long circulation in the bloodstream [72].

Recently, di- and triblock copolymers have emerged as micelle-forming long-circulating DDSs [73,74]. Self-assembling copolymers are commonly synthesized using anionic and ring-opening polymerization methods. These methods enable the variation in both molecular weight and hydrophilic-hydrophobic lengths. The variation of molecular weight of copolymers leads to achieve different physiochemical and biological characteristics. Longer hydrophobic block length induces the formation of a bigger hydrophobic core, thus increasing the ability to entrap hydrophobic drugs. In the case of the treatment of solid tumors, DDSs go through several important steps, such as circulation, accumulation, penetration, internalization, before releasing the drug in the target. Properties of DDS that determine their efficiency are surface activity, stability, targeting groups, morphology, and size. As reported, micelle with small sizes (e.g., 30 nm) has demonstrated more efficient penetration in tumor and higher anticancer activity of the drug [75].

These attractive characteristics have played a significant role in the design of new PNMs with a variety of structures and behaviors. Moreover, the designed PNMs have been successfully developed to release drugs through exogenous optical stimulation. Among different optical radiations, near-infrared (NIR) is more convenient for biological use due to near-infrared optical windows of biological tissues. NIR wavelength (980 nm) not only has lower toxicity than UV light to biological tissue but also can penetrate deeper in tissue. Furthermore, PNMs can be used simultaneously as drug delivery carriers and noninvasive optical imaging. For instance, Zhang et al. have synthesized new PNMs derived from the biodegradable poly(ε-caprolactone) derivatives, which have been coated with a fluorescent probe (Cy7) on the surface [76]. Kumar et al. reported the successful preparation of new PNMs loaded with NIR dye [Pt (II)-tetraphenyltetranaphthoporphyrin, Pt (TPNP)] [77]. Recently Wei’s group [78] have developed novel amphiphilic copolymer-based nanomicelles. The average diameter of these micelles was determined to be ~40 nm, which have demonstrated a relatively higher drug loading capacity of 70%. Moreover, authors have found that fluorescence quantum yield of micelles in water was higher, Φ = 22%. Besides, the half-maximal inhibitory concentration (IC_50_) of DOX-loaded nanomicelles against cancer cells was found to be 68.59 μg/mL while for DOX it was found to be 2.33 μg/mL.

##### Polyethers

Polyethers have been utilized as the core-forming element for encapsulating hydrophobic drugs. Polyethers are prepared through ring-opening anionic polymerization of alkenes to yield well-defined polymers with a low polydispersity index (PDI) and molecular weight (MW) distributions [79]. Poly(butylene oxide) have presented hydrophobic characteristics and, when integrated into block copolymers, can solubilize hydrophobic drugs [79]. PEO-PPO-PEO copolymers, which are known poloxamers, are usually employed as block copolymers for solubilizing hydrophobic drugs and synthesizing polymeric micelle formulations [80]. These block copolymers were the first applied for the delivery of the free drug in polymeric micelles.

##### Polyesters

Polyesters are other class of hydrophobic polymers which are regularly used in the preparation and design of polymeric micelles. The preparation of polyesters is generally carried out by ring-opening polymerization of cyclic esters and this synthetic method is known to yield high molecular weight polyesters with narrow polydispersity [81]. The main advantage of utilizing polyesters is their biodegradability [82]. The in vivo degradation strategy of the polyester backbone avoids the unrequired accumulation of the polymer in the body, thus decreasing the risk of chronic toxicity [82]. The block copolymers composed of the hydrophobic polyester block and hydrophilic block, such as PEG, were usually used to prepare micelle systems. For instance, micelle systems prepared using PCL-*b*-PEG-*b*-PCL displayed high loading up to 28% of paclitaxel [83].

##### Hydrophobic Poly(amino acid)s

Poly(amino acid)s have been utilized as hydrophobic core-forming blocks in amphiphilic block copolymers for solubilizing poorly-soluble drugs. Prepare of poly-amino acids is commonly carried out through living polymerization of α-amino acid N-carboxyanhydrides [84]. Commonly applied hydrophobic poly(amino acid)s are poly(β-benzyl-l-aspartate) (PBLA) and poly(γ-benzyl-α, l-glutamate) (PBLG). According to Thambi et al., PEG-*b*-PBLG bearing the disulfide bond (PEG-SS-PBLG) could solubilize poorly soluble camptothecin and form micelles in solution. The micelles displayed 20–125 nm size and the drug loading capacity was up to 12%. PEG-*b*-PBLA block copolymer was used to yield polymeric micelles for the physical encapsulation of doxorubicin [85]. The micelle preparation displayed 15–20% of doxorubicin loading and a 57–70 nm of size distribution.

##### Polyoxazolines and Polyoxazines

Recently, POx and POzi block copolymers have been applied for drug delivery applications and have shown strong potential as the materials for polymeric micelle drug carriers [86]. The synthesis of POx and POzi can be performed via LCRP strategy which results in carefully linear polymers of low molar mass distribution (PDI = M_w_/M_n_ from 1.01 to 1.3) and defined degrees of polymerization [59]. POx and POzi represent a versatile library of polymer structures. Depending on the 2-substitution of the 2-oxazoline or 2-oxazine monomers, the water-solubility of the producing copolymers range from highly hydrophilic MeOx or EtOx presented above to highly hydrophobic. Such structural variability makes easily accessible an expanded library of POx- and POzi-based block copolymers that can be utilized to yield polymeric micelle preparation of structurally diverse, poorly soluble drugs [87].

## 5. Preparation of Polymeric Micelles

The preparation of nanomicelles can vary based on the properties of the polymer chain length. Two protocols can be used for nanomicelles preparation included (1) direct dissolution and (2) solvent casting. Block copolymer that is moderately hydrophobic usually is self-assembled in nanomicelles through direct dissolution also known as a simple equilibrium method. In this method, the simultaneous dissolution of drug and copolymer with the appropriate ratio occurs in the aqueous solution which is then heated to initiate the formation of nanomicelles. During heating, the core of the structure undergoes dehydration that leads to the formation of nanomicelles. Solvent casting category can be classified into three techniques: (1) dialysis, (2) oil in water (*o*/*w*) emulsion, and (3) solution casting. The dialysis manner is convenient to nanomicelles from the nonwater-soluble copolymer. Drug and copolymer are dissolved in proper organic solvents miscible with water with a high boiling point. A solution of copolymer and drug are put into a dialysis bag and dialyzed against water for more than 12 h. During the dialysis, the organic solvent is slowly evaporated that initiates the formation of nanomicelles loaded with the drug. Although this method can overcome the issue to remove the organic solvent with a high boiling point, it has limitations such as drug lost due to low encapsulation efficiency. The second method in this category is the *o*/*w* emulsion that includes physical entrapment of components. The encapsulation procedure is based on the dissolution of drug and polymer in nonmiscible organic solvent with a small volume of water. The removal of the solvent via evaporation induces the physical entrapment of drug in the core of the formed nanomicelles [88]. The third method is called solution casting, which includes organic and aqueous solvents. Drug and polymers are dissolved in an organic solvent to obtain a transparent solution. Drug-loaded nanomicelles in the form of the thin film can be obtained after the removal of organic solvents under a high vacuum.

## 6. Biodistribution

The concept behind utilizing micellar carrier is to enhance the solubility and prolong the blood circulation time of a drug for great targetability and therapeutic benefits. Long circulation time depends upon biodistribution and metabolism. Since micelles are mainly administered via the intravenous route, the absorption phase is usually ignored. However, the in vivo biodistribution would influence physicochemical properties of the micelles, including their size and shape, core properties, surface modifications, and surface charge as well as targeting ligand functionalization [89]. In general, spherical micelles of ~3–5 nm size are less probably to be taken up by the macrophages due to minimum surface area and are excreted by the kidney [90]. Small particle size display broad distributions as they can simply cross tight endothelial junctions to enter into extravascular extracellular space (EES). However, the large size of nanoparticles (≥10 nm) is rapidly cleared by RES via spleen and fenestra of the liver [91].

The polymeric nanomicelles possess the merit of the EPR effect due to their small size [92] where they passively diffuse through the endothelial lining and remain at the site of inflammation owing to poor lymphatic drainage, thus releasing their payload at the specific site. However, despite their stability of curcumin loaded polymeric micelles (size ~80 nm) synthesized using ω-methoxy poly (ethylene glycol)-*b*-(N-(2-benzoyloxypropyl) methacrylamide) (mPEG-HPMA-Bz) undermine their advantage by the EPR effect. Moreover, this impact was seen in other formulations of curcumin, namely, LDL (30 nm), liposomes (180 nm) and intralipid (280 nm). The natural properties of drug molecules and not just aromatic groups (as shown in the case of PTX) are responsible for their performance [93].

The micellar biodistribution is highly influenced by the size, shape, and surface charge. Compared to neutral and negatively charged micelles, positively charged micelles are readily taken up by the RES due to increased interaction with the negatively charged cell membrane [90]; however, negatively charged micelles decrease the rate of cellular uptake and improve the blood circulation time. When two charged micelles of Tyr-PEG/PDLLA (neutral) and Tyr-Glu-PEG/PDLLA (anionic) were tested in mice for pharmacokinetics [94], both displayed similar blood clearance kinetics, but anionic micelles showed ~10 times lower biodistribution in liver and spleen compared with neutral micelles, due to synergistic steric and electrostatic repulsion.

The impact of core on biodistribution is highly variable and depends upon the core composition and its stability. For example, when PTX-loaded pluronic P105 micelles were compared with mixed micelles composed of a similar copolymer with just the addition of hydrophobic lamella, a strong decrease in clearance was observed [95]. This probably due to the addition of hydrophobic lamella that might have enhanced hydrophobic interaction in the core, stabilizing the micelles, thereby reducing its uptake by the liver.

## 7. Nanomicelles Delivering Anticancer Drugs

Liposomes and polymeric micelles, representing nanoscale medication distribution methods, have been advanced as a crucial tool for cancer treatment [96,97]. As a result, appropriate transmission time, tumor concentration, and weakening accumulation in potentially risked healthy organs and tissues could be achieved for administered anticancer drugs.

Nevertheless, liposomes application and lipid-based drug delivery systems have been narrowed due to attained drug resistance and poor targeting [98,99]. DOX-loaded micelles, described as DDSs by Kataoka’s group in the early 1990s, have been utilized for the distribution of various anticancer agents in clinical and preclinical investigations [100]. PNMs employ passive targeting, folate, pH-sensitive, and thermosensitive DDSs to beat drug resistance through perfect targeting (Figure 4) [101,102,103,104].

Low absorption, bioavailability, and drug aggregation are considered major anxieties in therapeutic presentations representing another considerable drawback of anticancer agents, i.e., poor water solubility [106]. In turn, PNMs can rise considerably the anticancer medication water solubility from 10 to 5000 fold [107] and contain an inner core consisting of a hydrophobic copolymer, where hydrophobic drugs can be entrapped. Additionally, the outer shell of hydrophilic copolymer contained in PNMs decreases the relations of drugs with the aqueous medium and keeps them steady [108,109].

A different class of “conjugate” was established in later studies using PEG-b-P(Asp) and PEG-b-P(Glu), which can create a stable complex with transition metal complexes-based drugs such as cis-dichlorodiammine platinum(II) (cisplatin), dichloro(1,2-diaminocyclohexane)platinum(II) (DACHPt) and cis-oxalato-(trans-l)-12-diaminocyclohexane-platinum(II) (oxaliplatin) [110]. In this case, the development of the block copolymer micelle is motivated by the complexation of the polyacid component with the metal of the drug molecule. The complex of PEG-b-P(Asp) and cisplatin spontaneously self-assembled into polymeric micelles with a very small size distribution. The drug was released from the micelles via the exchange of ligands with chloride ions in biological environments [111]. The polymeric micelles (PEG-*b*-P(Glu)) containing another platinum drug DACHPt were reported by Kataoka’s group [112]. Similarly, to cisplatin, DACHPt was bound to the carboxylic groups of P(Glu) polymer block through coordination bonding of platinum and the selected drug was released via ligand exchange of DACHPt with chloride ions in the environment.

Lo et al. developed several graft copolymers and investigated their self-assembly to form core-shell nanomicelles. The prepared nanomicelles were further encapsulated with 5-fluorouracil (5-FU) and tested for structural deformation and drug release performance to evaluate their applicability for intracellular drug delivery [113].

In another study, there was a dual responsive biocompatible nanocarrier with appropriate paclitaxel drug loading and controlled release efficiency. The authors determined the size of the prepared nanocarrier and the encapsulation efficiency to be ~100–230 nm and 98%, respectively. It was found that nanocarrier formed aggregates under slightly acidic pH (pH 6.9). Rapid release of PTX at high temperature (37 °C) and low pH (4, 6.8, 7.2) was compared to a lower temperature (20 °C) and pH (4). These findings indicated that nanocarriers would accumulate and release the drug selectively in cancer cell tissues to achieve targeted delivery of anticancer drugs [114].

Similarly, novel copolymer micelle-based nanocarriers were developed using dialysis method. Nanomicelles formed showed temperature and pH responsive release performance of methotrexate (MTX) drug. The developed nanocarriers would be a promising site-specific drug delivery system which could improve the drug accumulation in pathological sites. The vitro cytotoxicity, cellular uptake and in vivo antitumor activity studies would further confirm nanomicelles ability to respond to changes in pH and temperature [115].

## 8. Targeting Nanomicelles

Target moieties can be paired to generate dynamic targeting nanomicelles, for maximizing distribution and minimizing side effects. These active nanomicelles can target cells based on (1) relationships with particular targets and (2) conjugation with locally functioning signal protein [116]. In addition, paclitaxel-loaded phosphatidyl ethanolamine (PEG-PE) micelles, adjusted with MCF-7-selective phage fusion proteins targeting tumor cells were advanced by Tao Wang et al. [117]. As a result of the study, greater tumor selectivity in cancer cells was observed in the targeted phage nanomicelles rather than in normal cells. Consequently, this targeting method can boost the anticancer outcomes of paclitaxel in mice. Furthermore, alternative target agents such as antibody [118,119,120], peptides [121,122,123,124], and aptamer [125,126] were studied. Interestingly, to target pancreatic cancer, Ahn et al. have organized new nanomicelles conjugated antitissue factor antibody (TF) burdened with platinum drugs [119]. These nanomicelles demonstrate 15-fold greater cellular uptake and substantial anticancer outcome for extra than 40 days in pancreatic cancer than nontargeted micelles. Recently, Sarkar et al. [127] developed stearic-g-polyethyleneimine acid amphiphilic nanomicelles functionalized with folic acid-based carbon dots (CDs) for targeted anticancer drug (DOX) delivery and concurrent bioimaging for triple negative breast cancer (TNBC). The fluorescence property offered by folic acid derived CD allowed CDSP-25 to act as a promising bioimaging tool for TNBC (Figure 5).

## 9. Stimuli-Responsive Nanomicelles

Stimuli-responsive nanomicelles and their perspective as efficient delivery systems for a specific site have been actively discussed in recent studies on drug delivery technology. These nanomicelles were practiced in the drug delivery field and benefitted substantially to pharmaceutical investigation [128]. Despite the internal and external application of so-called “smart” or “environmental-sensitive” nanomicelles, they exhibit an exceptional feature of modifying physical structure in response to small environmental changes [129]. The stimulus types, administering the arrangement of stimuli-responsive nanomicelles, which is mandatory to act on the micelles to exert such required stimulus-outcome as physical, chemical, or biological [130]. Moreover, the chemical configuration of stimuli-responsive nanomicelles may be changed intensely reacting to pH, temperature, light, ultrasound, and enzyme deprivation (Figure 6) [131,132].

### 9.1. pH-Sensitive

For a long time, targeted and stimuli-responsive nanomicelles for drug delivery have been employing changing pH values of the human body. This is primarily accomplished by two methods. First, through nanomicelles comprising of ionizable groups, which experience solubility changes with environmental pH alterations. Second, by the utilization of nanomicelles with acid-sensitive bonds, whose hydrolysis leads to the discharge of the drug from the nanomicelles core [133,134]. Among all pH-responsive manners, gastrointestinal products, employing pH-sensitive coatings are considered as the most frequently employed ones.

Normal tissues are considered less acidic compared to the pH of the tumor environment, which makes up 6.5–7.2. These pH dissimilarities are used to cleave the acidic-sensitive molecules in order to release therapeutic molecules from the nanomicelles. Briefly, drug molecules are trapped inside nanomicelles in the physiological pH 7.4. Acidic circumstances like a tumor cell, endosomes, or lysosomes may lead therapeutic agents to be released selectively.

**Figure 6 nanomaterials-11-00070-f006:**
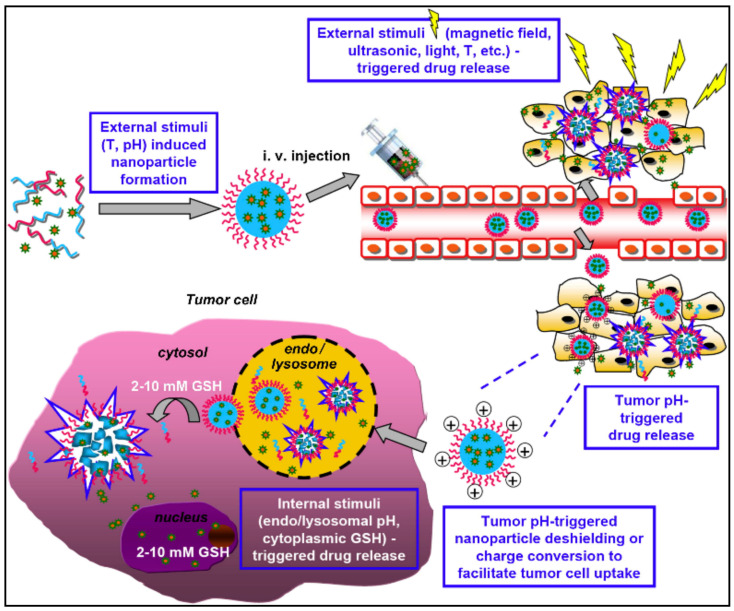
Emerging drug delivery systems based on stimuli responsive polymeric nanomicelles (i) formation of micelles through pH and temperature external stimulus application; (ii) drug release; (iii) enhancing cell uptake of drugs and release under acidic effect; and (iv) strong redox potential is used to increase intracellular drug release inside cancer cells. Reprinted with permission from [135]. Copyright © 2013, Elsevier.

Up to date, several different drug release mechanisms and pH-sensitive micellar drug delivery systems can be summarized as the following:

#### 9.1.1. Drug Release-Based Protonation or Deprotonation of Polymers

Amphiphilic copolymers consist of hydrophilic and hydrophobic parts. In this approach, the properties of the amphiphilic polymer with ionizable groups can be changed in response to the changing pH of the environment. Acidic pH caused protonation leads to hydrophilic-hydrophobic phase transition for the anionic polymers, and hydrophobic- hydrophilic phase transition for cationic polymers, inducing deformation of amphiphilic structures leading to drug release [136]. Anionic polymers with free carboxylic acid groups are generally utilized as pH-sensitive copolymers, such as poly (acrylic acid), poly (methacrylic acid) (PMAA), poly (2-ethyl acrylic acid) and poly (glutamic acid) [137]. The carboxyl groups are deprotonated to exhibit hydrophilicity at neutral pH; however, they become protonated and hydrophobic at acidic pH. For example, a promising amphiphilic block copolymer consisting of PEG-poly polymers utilized to encapsulate paclitaxel for anticancer studies [138]. The synthesized polymers have consisted of PEG as the hydrophilic part and the 4-phenyl-1-butanol modified poly(aspartate) as the hydrophobic part. Paclitaxel was incorporated into the micelle core though the self-association method. The unmodified carboxyl groups of poly (aspartate) polymer were deprotonated at physiological pH to keep the stability of the micelle system; however, they were protonated and could induce the release of paclitaxel at acidic pH.

Moreover, amphiphilic polymers with ionizable groups such as tertiary amine, imidazole, and pyridine groups are usually used as an element to develop the amphiphilic copolymers. These amino-rich hydrophobic elements may be deprotonated to keep hydrophobic at physiological pH but can be protonated to become hydrophilic at acidic pH. In this technique, the hydrophilicity of the micelles is destroyed, leading to the disaggregation of the micelles resulting in drug release. Histidine was applied as the stone group for pH-sensitive micelles because of its ability to protonate at acidic cancer cell pH while remains neutral in physiological pH [139].

#### 9.1.2. Drugs Release-Based the Separation of the Amphiphilic Block Micelles

The hydrophilic and hydrophobic components of the amphiphilic copolymers are linked by pH-sensitive bonds to design pH-responsive targeting delivery system with encapsulated anticancer drugs. The pH-sensitive linkers on these copolymers are commonly stable at pH 7.4, but are hydrolyzed in acidic environment, whereby the micelles are disaggregated to provide fast drug release.

Ma et al. prepared amphiphilic dextran-retinal (DR) by combining all-trans retinoic acid and dextran through a hydrazone bond, followed by self-assembly to produce pH-sensitive DR micelles [140]. The DR micelles could be encapsulating DOX into the hydrophobic core to yield DOX-loaded DR micelles. The release of DOX from DOX-loaded micelles was accelerated under acidic environment due to the cleavage of acid labile hydrazone bond. The in vitro release of DOX showed that less than 10% of DOX was released in PBS at pH 7.4, but remarkably high at pH 5.0 with about 100% of DOX release within 24 h. In vitro anticancer test of DOX-loaded micelles, including apoptosis, cell cycle, and cellular senescence have been investigated in human breast tumors. Additionally, bidistribution and anticancer effects were tested in MCF-7 xenografted mouse models. Compared with free DOX, DOX-loaded micelles improved cancer inhibition efficiency and reduced toxicity. Huang and coworkers developed pH-sensitive poly (ethyleneglycol)-imine-benzoic-dipalmitate (PEG-I-dC16) amphiphilic polymers for DOX delivery [141]. Acid-sensitive Schiff base groups have been utilized to link PEG and double-stranded C16 chains to prepare the PEG-I-dC16 micelles, by comparison with the PEG-A-dC16 micelles bonded by nonacid sensitive amide bonds, the rapid release of DOX from the PEG-I-dC16 micelles was accelerated by changing the pH from 7.4 to 6.5. Confocal microscopy technique displayed that the acid-labile micelles had more accumulation in cellular than nonacid-labile micelles and free DOX, and the cytotoxicity of DOX-loaded acid-labile micelles was greater than that of DOX-loaded nonacid-labile micelles against A549 and HepG2 cancer cells.

#### 9.1.3. Drug Release-Based the Reduced Hydrophobicity of the Polymeric Micelles

The integration of 2,4,6-trimethoxybenzaldehyde to the hydrophobic part via acetal bonds is the most common strategy in this approach. Under the acidic pH environment, the acetal hydrolysis resulted in strong micelles swelling, because of decreasing the hydrophobicity of the hydrophobic part. Zhong and coworkers studied the synthesis of pH-sensitive biodegradable micelles, which were designed from the block copolymers PEG-PTMBPEC [142]. The change of the micelles size in response to acetal hydrolysis was confirmed by dynamic light scattering (DLS). The results demonstrated that putting micelles into pH 4.0 acetate buffer (0.5 M) resulted in fast and significant micelles swelling instead of micelle disruption. The size of micelle increased from 150 nm to about 400 nm in 3 h, reaching over 1000 nm after 12 h. However, no change of micelle size was seen over two days at pH 7.4 at the same buffer concentration. The micelles have been incubated with carboxy fluorescein and Nile red after acetal hydrolysis and then seen colocalization of carboxy fluorescein and Nile red, confirming the amphiphilic micelles. Moreover, the ^1^H NMR studies on micelles before and after acetal hydrolysis further corroborated swelling of the micelles. In vitro release studies displayed a remarkably faster release of paclitaxel and DOX at pH 4.0 and 5.0 than at pH 7.4.

Gu and coworkers constructed PEG-PH-PLLA nanomicelles based on self-assembly of triblock copolymers poly(ethylene glycol)-poly(Lhistidine)-poly(l-lactide) [143]. The anticancer drug DOX was encapsulated in the nanoparticles. The self-assembly nanoparticles were divided into three layers, including hydrophobic PLLA part, pH-sensitive PH blocks and hydrophilic PEG chains. The PH layer swelled with the protonation/deprotonation at various pH to monitor the DOX release. After 24.5 h, the accumulated release percentage of the nanoparticles at pH 5.0 was approximately 80%, while that of nanoparticles in pH 7.4 was less than 40%. In vitro studies in HepG2 cells displayed that the anticancer effect of DOX-loaded nanoparticles was preferable to free DOX.

#### 9.1.4. Drug Release-Based the Rupture of the Acid-Labile Bond between the Drug and Polymer

In these delivery systems, the amphiphilic polymers are commonly stable in blood circulation and acid-sensitive bonds are hydrolyzed after cellular internalization. In this technique, the micelles are not depolymerized, and the drug release is relatively slow. For instance, hydrazone bond has been successfully used to integrate DOX into a broad variety of amphiphilic polymers. A novel class of self-assembling amphiphilic blocks, poly(ethylene glycol)-poly(aspartate hydrazone adriamycin) (PEG-P(Asp-Hyd-ADR), was specifically developed and prepared by conjugating adriamycin (ADR) to the side chain of the core-forming PAsp competent via the pH-sensitive hydrazone bond [144]. In vitro release results displayed that the micelles release ADR decreases depending on pH values from pH 7.4 to 3.0. CLSM reveals that the micelles are trapped in lysosomes, where they are programmed to function by responding to low pH, and the released ADR accumulates in the cell nuclei and successfully suppresses the synchronizing cell viability of cancer cells.

Zhong et al. developed endosomal pH-sensitive paclitaxel (PTX) micelles by encapsulating PTX onto PEG-PAA copolymers via acid-labile acetal bond to the PAA block and tested for growth inhibition of human cancer cells [145]. The drug release behavior in vitro displayed that drug release from PTX nanoparticles was strongly pH-dependent, in which 86.9%, 66.4%, and 29.0% of PTX were released from PTX drug at 37 °C in 48 h at pH 5.0, 6.0, and pH 7.4, respectively. Moreover, MTT test exhibited that these pH-sensitive PTX displayed high anticancer effect on KB and HeLa cells as well as PTX-resistant A549 cells.

#### 9.1.5. Drug Release-Based Other pH-Sensitive Mechanism

Various pH-response nanomicelles are advanced with the concept of good biocompatibility and strong selectivity [146,147,148,149,150,151,152,153,154,155,156,157,158]. For instance, pH-sensitive poly(ethylene glycol)-poly(aspartate hydrazone driamycin)loaded nanomicelles, developed by Bae et al., discharge selectively in the strong acidic medium because of pH-sensitive hydrazine group [159]. Recent research has employed the change of systemic pH values and exploit fluctuating of gastrointestinal for drug delivery to intestinal and colon. Xu et al. [160] reported a new nanomicelle with dual-pH-response for axitinib and doxorubicin delivery to cancers. This method is based on conjugating doxorubicin to *N*-(2-hydroxypropyl) methacrylamide (HPMA) via an acidic-sensitive hydrazone group with axitinib, subsequently compressed onto the nanomicelles form. Collapsing at cancer cell (pH 6.4) benzoic-imine bond was utilized as a carrier, where the axitinib was released. Afterward, the nanomicelles were endocytosed leading to doxorubicin discharge by acid cleavage of the hydrazone connections inside the lysosomes (pH 5.0). The advanced nanomicelle system led to effective tumor targeting with inhibition of the tumor development by 88%. Recently, Wu et al. [161] introduced a new nanomicelle (20–30 nm) based on the chemical modification of hydroxyapatite for pH-responsive DOX delivery (Figure 7).

Amphiphilic polysaccharides are familiarized as outstanding materials for biomedical claims due to low toxicity, biodegradability, and excellent bioactive features [162,163]. Modified heparin and histidine (HDH)-based pH-sensitive micelles (size ~110 ± 12.36 nm) were defined by Tsai’s group in detail [164]. The drug loading capacity made up 14.52% and encapsulation effectiveness accounted for 65.47%. Remarkably, at pH 5.0 and pH 6.0 about 91% and 63% of zinc phthalocyanine drugs were released after 96 h. Additionally, according to phototoxicity outcomes, free ZnPc micelles had lower toxicity compared to ZnPc-loaded at identical concentrations, representing an encouraging chance of HDH for ZnPc drug with simultaneous advancement of PDT efficiency.

Likewise, the novel delivery method for DOX using glycol-polymeric micelles (MH-S-S-AcMH) was established by Cheng et al. [155]. Particularly, 20 nm average-sized amphiphilic MH-S-S-AcMH were able to self-assemble to micelles. Provided weakly acidic environment (pH = 5), the fast release of DOX could be observed. Moreover, DOX-encapsulated micelles revealed improved cytotoxicity to HeLa cells than bare DOX. Consequently, these outcomes pointed out the efficient utilization of DOX-encapsulated MH-S-S-AcMH for drug delivery development and improving anticancer treatment.

Presenting pH-responsive amphiphilic polymers-based micelles that control drug delivery and release, characterizes an effective approach when scheming pH-responsive NPs. For example, synthesized novel nonionic micelles derived from alginate that might be utilized upconversion nanoparticles encapsulated simultaneously with doxorubicin were presented by our group [17]. The encapsulation effectiveness accounted for 81.2% whereas drug loading capacity made up 18.3% for the developed nanomicelles (Figure 8A). In addition, 90% of DOX was discharged from nanomicelles at pH 5.0 that could be caused by weak ionic interaction between nonionic nanomicelles and DOX. It is noteworthy that KB cancer cells received effectively DOX-loaded nonionic alginate micelle through folate-mediated endocytosis, resulting in pH-responsive DOX discharge and consequent apoptosis, Figure 8B.

Numerous strategies to develop effective biodegradable nanomicelles for bioapplications have been elaborated so far. For example, polyacetal dendrimers were synthesized with β-cyclodextrin and adamantane-conjugated zwitterionic poly(sulfobetaine) by Huang et al., [165] leading to supramolecular amphiphilic LDBCs being formed by host-guest recognition (Figure 9). Particularly, LDBCs were able to self-assemble into pH-responsive and biodegradable nanomicelles by changing the hydrophobic/hydrophilic ratio, which represented exceptional resistance to protein absorption and worthy of biocompatibility.

Because DOX-loaded nanocarriers demonstrated an effective pH-responsive drug discharge strategy, DOX might be encapsulated into the nanomicelles. Moreover, these nanocarriers revealed higher internalization effectiveness and notable cytotoxicity against cancer cells compared to conventional PEG-coated nanocarriers. An alternative study has been carried out on a unique type of hyperbranched amphiphilic copolymer synthesis for smart drug delivery [166]. These novel copolymers could be self-assembled into nanomicelles with diameters varying from 70 to 100 nm and were studied for the discharge of DOX. Under acidic conditions, DOX-loaded nanomicelles (~90 nm) with a capacity of 8.2 wt% demonstrated a more efficient drug release. Besides, the observation confirms that DOX-loaded nanomicelles displayed notable cytotoxicity against HeLa cells. Observations reveal that the above developed polymeric nanomicelles may be considered as a favorable pH-responsive carrier for cancer treatment.

Quercetin-chitosan conjugate (QT-CS) self-assembled into micelles by Mu et al. [167] was able to encapsulate DOX at a high ratio with the minor particle diameter of 136.9 nm and zeta potential of +16.2 mV. These nanomicelles demonstrated a strong-release profile in the gastrointestinal environment (pH 1.2/pH 7.4). In addition, QT-CS nanomicelles could encourage cellular uptake of doxorubicin that was 2.2-fold higher compared to free drug.

Recently, Kou et al. [168] established a novel amphiphilic starch-based polymer (R-St-mPEG), with good self-assembly property into spherical micelles (~200 nm, CMC ~0.039 mg mL^−1^). Curcumin drug could be facilely encapsulated into the nanomicelles. The bioimaging release study demonstrates that the release of curcumin from the nanomicelles is a slow, stable, and pH-responsive. Similarly, smart micelles-based modified lignin for ibuprofen (IBU) delivery has been proposed by Cheng and coworkers [169]. The results showed that the IBU drug was released from the core of the developed micelles within 72 h in an acidic environment (Figure 9).

**Figure 9 nanomaterials-11-00070-f009:**
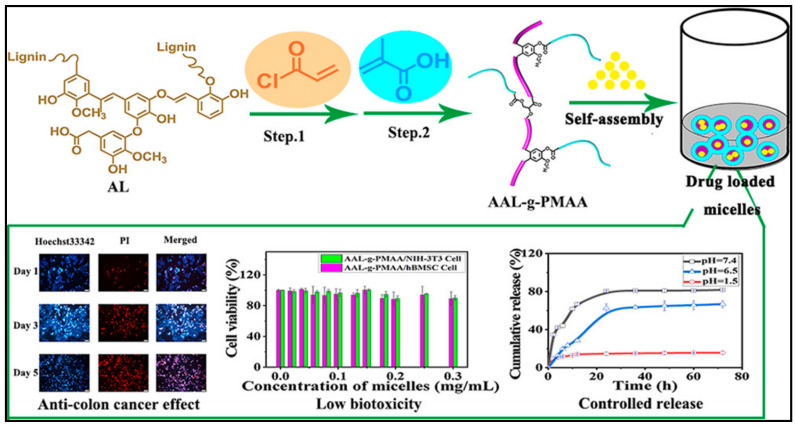
Structure of AAL-g-PMAA and self-assembly into nanomicelles and ibuprofen (IBU) can be loaded in the Core (up). Drug release profile, cytotoxicity study and bioimaging results (down). Reprinted with permission from [169]. Copyright © 2020, American Chemical Society.

### 9.2. Temperature Sensitive

Nowadays, bioactive distribution to the human organism is effectively carried out by temperature-modulated polymeric systems. The current feature is accomplished with the help of nonlinear, sharp, and irregular modification of the characteristics of delivery system constituents as a reaction to temperature rise and can frequently be observed in liposomes, polymer micelles, and nanoparticles because of low critical solution temperature. CMC may be significantly influenced by temperature fluctuations. For example, *N*-isopropyl acrylamide, known as the most frequently utilized polymer, can transform to a hydrophobic polymer from the hydrophilic polymer at approximately 32 °C. However, Soga et al. have discovered different temperature-sensitive polymeric micelles [170,171,172]. Additionally, thermosensitive nanomicelles, developed by Chung et al., utilized polybutyl methacrylate (PBMA) as a hydrophobic group whereas poly *N*-isopropyl amide was employed as a temperature-sensitive hydrophilic part [173]. As a result, the release of DOX from pNIPAM-b-PBMA-based nanomicelles made of 15% after heating for 15 h at 30 °C, compared to 90% drug release at 37 °C. According to cytotoxicity tests, less than 5% of cells died at 29 °C, while this was increased to 65% at 37 °C due to the temperature-responsive drug release.

Temperature and pH-sensitive drug carriers based on pseudopolypeptide micelles were developed by Chen et al. (2018) employing a self-assembly approach with 11% DOX-conjugating fraction [174]. These micelles possess the property of unchanging sphere-shaped morphology and approximately 123 nm of the hydrodynamic dimension. In addition, the micelles exhibited thermosensitive performance for DOX release. According to cellular uptake investigation, micelles were able to move into the cells via caveolae-mediated endocytosis. Similarly, Ghamkhari et al. [175] developed biodegradable and spherical nanomicelles with an average size of about 25 nm and 45 ± 10 nm for the enhancement of docetaxel (DTX)-loading. The DTX-loading efficiency was determined to be 95.5%. The release of DTX from the nanomicelles demonstrated temperature-responsive manner.

In another good example, [176]. Novel polymeric materials were self-assembly into nanomicelles with a core-shell structure and encapsulated with a hydrophobic Pt (IV) complex and Fe_3_O_4_ nanoparticles. The cytotoxic of platinum drug-loaded nanomicelles on a head and neck cancer cell (SQ20B) were studied in vitro. Moreover, taking the advantage of magnetic functionality nanomicelles in the presence of a near-infrared fluorescent dye, magnetically targeting them to a cancer site in a live animal xenografted model was demonstrated. Similarly, very recently, Drage et al. [177] developed DOX-mPPD micelles based on temperature-sensitive copolymers with good loading capacity, 6.79% and release efficiency of 60.27% (Figure 10).

### 9.3. Light Sensitive

The specificity of light irradiation in terms of serving as an external stimulus for drug release is determined by its accurate temporal and special control. Optical windows in near-infrared (NIR) region in which light has its deepest penetration in tissue cause minimum damage to cells [178]. This enables the wide application of NIR irradiation for targeted drug delivery, which has been explored in recent years [179,180].

Nowadays, irradiation-induced reversing of the solubility state of nanomicelles from hydrophobic to hydrophilic state and vice versa is becoming popular. Therefore, this technique is commonly applied for controlled drug release [181,182]. For instance, IR light-caused release of a fluorescent probe like Nile Red from the micellar system was reported by Andrew et al. [183]. In addition, the novel product of modification of polypyrrole with PEG and pyrene-oxabicycleheptenealkyne (POA) was designed recently to link covalently hydrophobic anticancer therapeutic units [184]. PPy aimed to transfer NIR irradiation which is supposed to release drugs by retro D-A reaction.

Another approach was developed by Li and coworkers whereby block copolymer nanomicelles consisting of poly(oligo(ethylene glycol)methacrylate)-block-poly(furfuryl methacrylate) (POEGMA-b-PFMA) possessed thermosensitive properties and could deliver drug molecules according to NIR response [185]. Such PNMs bearing NIR dye indocyanine green (ICG) and DOX could efficiently and selectively deliver the drug to the targeted site. When encapsulated with ICG and DOX, the PNMs could deliver the drug efficiently. More specifically, the photothermal effect resulted from 805 nm-NIR irradiation of ICG, which is located in the internal side of micelles caused an increase in temperature in the micellar core. This, in turn, led to the accelerated release of DOX. Likewise, a pH-responsive multiblock amphiphilic copolymer with galactose-targeted moiety was recently developed, which also incorporated NIR dye indocyanine green probe and therapeutic unit and demonstrated high uptake specifically to HepG2 cells [186]. Thus, the development of NIR-responsive PNMs is encouraged due to clear advantages of controlled drug release that enhances the efficiency of anticancer therapy yet diminishes the following side effects.

Recently, Pourjavadi et al. [187] developed amphiphilic micelles based on the chemical modification of chitosan biopolymer through conjugation with poly(acrylamide) and poly(N-isopropyl acrylamide). The photoresponsiveness futures of the prepared micelles were achieved through complexation with the gold nanorods. The drug release experiments displayed that about 38% of paclitaxel drug was released under NIR light irradiation (Figure 11).

Pan et al. [188] recently studied the fabrication of smart nanomicelles produced from the self-assembly of IR825 with methoxy poly(ethylene glycol)-block-poly(l-aspartic acid sodium salt) (PEG-PLD-IR825). Studies showed that these nanomicelles possess good drug loading efficiency (~21%) while in vivo studies demonstrated their long circulation half-life, suitable EPR effect, a minor release of cyanine, and notable accumulation in the mitochondrion. The latter bolstered significant photothermal response upon NIR irradiation at a concentration of drug as low as 50 µg/mL, which was revealed in vitro studies in HeLa cells. The only drawback observed was the tendency of IR825 to photobleaching when color disappeared during the laser irradiation in sample solutions.

Recently, light-responsive polymeric micelle (3PEG–PCL) with good stability and low CMC values has been developed (Figure 12A) [189]. Remarkable, UV radiation-caused fast DOX release and cytotoxic study displayed that DOX could be accumulated within the cells under normal tumor conditions, Figure 12B. Similarly, Yang et al. prepared a novel polymeric nanocarrier for bioimaging and drug delivery (PEG-g-p(GEDA-co-DMDEA) [190]. In one more study, Wang et al. reported the integration of 1,2-dimyristoyl-Sn-glycerol-3-phosphocholine with cyanine dye and self-assembled into micelles [191]. Similar nanomicelles-based on upconversion, and poly(phosphorylcholine)for DOX delivery have been developed recently [192,193].

Wang et al. prepared a series of new rodlike nanomicelles based on poly(ethylene oxide)-block-poly(sodium acrylate) [194]. These nanomicelles displayed high NIR emission and strong stability. As demonstrated by in vivo NIR imaging result, the rodlike nanomicelles can successfully stain cancer sites with a long retention time. Moreover, the nanomicelles exhibited effective anticancer efficiency by accurately killing cancer cells without affecting healthy organs in vivo.

Chen et al. prepared a novel NIR-II/PA multimodal probe based on the CH1055 derivative [195]. This bioimaging probe was encapsulated into amphiphilic PEGylated phospholipids (DSPE-PEG5000-NH_2_) in aqueous solution, then integrated with an engineered small protein anti-EGFR Affibody which can enhance the targeting efficiency for cancer cells. Affibody-DAP can bind EGFR-positive tumors in FTC-133 subcutaneous mice model with excellent PA and NIR-II fluorescence signals (Figure 13a). Recently, Sun et al. developed an effective and high-performance dye based on a DD–A–DD scaffold. By using the dialkyl substituted fluorene part between the second donor (diphenylamine) and thiophene, the novel designed NIR-II dye displayed outstanding performance in delineating tumor sites with both NIR-II/PA modes (Figure 13b).

Most recent conjugated polymer probes based on NIR-II fluorescence which naturally induce interference from background fluorescence. Fan’s group described a new method to construct a smart NIR-II probe using the bioerasable intermolecular donor–acceptor interaction [196]. The donor semiconducting polymer and acceptor nonfullerene (ITTC) were accurately coated with an amphiphilic polymer in aqueous solution. The fluorescence of conjugated polymers was quenched likely due to the strong doner–acceptor interaction. It was proposed that hypochlorite can degrade the acceptor ITTC (Figure 13c). After that, the interaction disappeared, and the NIR-II fluorescence restored. This strategy formed new opportunities to explore NIR-II imaging with activatable probes in vivo. An activatable and biocompatible NIR-II nanoprobe that particularly effective in tumor redox and acid microenvironment was developed by Pu’s group. The method has broad prospects in stimuli-activatable NIR-II photo-nanotheranostics (Figure 13d) [197].

Most recently, NIR-II imaging probes which can be utilized in renal impairment (RI) imaging and excrete through renal clearance are very rare. Pu and coworkers studied a new renal-clearable molecular semiconductor (CDIR2) that can real-time monitor kidney dysfunction in NIR-II components (Figure 13e), a NIR-II fluorophore with classic D-A-D structure (λem = 1050 nm) and a renal-clearance-enabling segment (HPbCD). In vivo study, they observed CDIR2 has greater SBR than traditional NIR molecules (ICG or CCD), and has a high renal clearance efficiency with excellent biocompatibility. Furthermore, CDIR2 has potential to use in noninvasive monitoring of kidney dysfunction and nephrotoxicity and high-throughput drug screening with the animals alive [198].

Effective monitoring the drug release in NIR-II biochannel has extensively attracted researchers’ attention. Recently, Fan reported a new method to overcome this problem by introducing a semiconducting nanoprobe for NIR-II/PA imaging-guided photothermal initiated NO/photothermal therapy. Semiconducting nanoprobes, containing a donor (fluorene and thiophene) and an acceptor (diketopyrrolopyrrole, DPP), were mixed with SNO-based NO-donor coating with amphipathic polymers, to form core–shell NPs [199]. As expected, NO can be released to achieve therapeutic effect under continuous laser excitation at 808 nm. Moreover, the NPs showed outstanding biocompatibility, no body weight loss was detected in all groups (Figure 13f). It has been well known that Pt (II) metallacycles-based supramolecular coordination complexes (SCCs) can act as an antcancer compound, but poor photostability, low tumor uptake and penetration depth limited in vivo applications. Recently, Stang and Sun developed a new method by assembling Pt (II) metallacycles based SCCs with a NIR-II dye (SY1100) in the presence of DSPE-PEG5000 as a carrier [200]. Compared with cisplatin, this NIR-II theranostic agent showed good efficiency of tumor growth inhibition and better biocompatibility (Figure 13g).

**Figure 13 nanomaterials-11-00070-f013:**
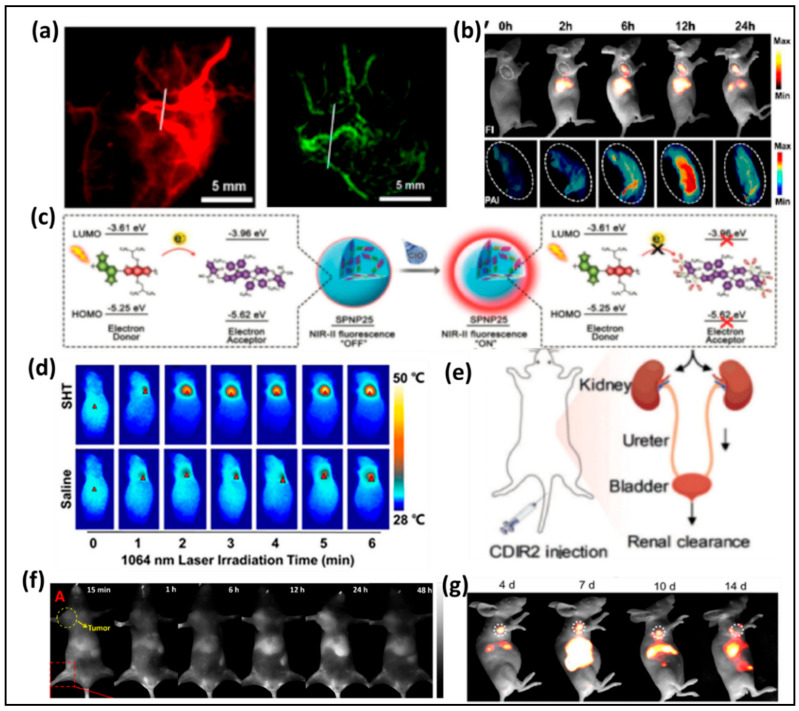
(**a**) NIR-II and PA imaging of blood vessels using Affibody–DAPs, (**b**) NIR-II and PA imaging of SY1080 NPS, (**c**) The mechanism of bioerasable intermolecular donor–acceptor interaction of SPNP25 with ClO^−^, (**d**) IR thermal images of living mice with 4T1 tumors under 1064 nm laser excitation (1 W cm^−2^) at 48 h, (**e**) Diagram of the renal excretion of CDIR2 through the urinary system, (**f**,**g**) NIR-II images of monitoring the Pt(II) metallacycle-based NIR-II theranostic nanoprobe’s therapeutic response in HepG2 tumors. Reprinted with permission from [201]. Copyright © 2020, Elsevier.

In contrast, the application of high-energy UV radiation is limited because of inadequate tissue penetration and harmful effect on the organism. In some studies, a novel amphiphile of hyaluronan-o-nitrobenzyl-stearyl was self-assembled into nanomicelles (HA-NB-SC) and doxorubicin (DOX) encapsulated within the core [201]. The diameter of DOX-loaded nanomicelles was determined to be ~139 nm. The results displayed that the HA-NB-SC nanomicelles can be easily taken up by HeLa cells and could inhibit the proliferation via recognition of the HA unit. The release of DOX from the developed nanomicelles at targeted sites could be taken apart upon UV light (365 nm).

### 9.4. Ultrasound Responsive

The ability of ultrasound to produce a frequency of about 20 kHz is proposed to use for drug uptake. Several works have been reported thus far on increasing drug delivery under ultrasound because of improved tissue penetration, correlation the features of cell membrane perturbation, and drug release [202,203,204,205]. For example, DOX release and its intracellular uptake from such popular ultrasound-responsive nanomicelles as pluronic micelles were studied by Marin et al. The study showed increased DOX release under high-frequency ultrasound [206].

Another approach has been developed by Pruitt and Pitt in which DOX stabilized with PEG-phospholipid was encapsulated in ultrasound-sensitive nanomicelles [207]. In vivo studies revealed that the release of DOX was significantly increased upon treatment with ultrasound that, as a result, retarded tumor growth higher than without treatment.

The biodistribution under the ultrasound effect of nanomicelles with the varying sizes around 12.9 nm (fluorescently labeled Pluronic micelles) loaded with DOX was studied by Gao and coworkers in ovarian cancer [208]. Pluronic (unstabilized nanomicelles) and their mixture with PEG-dia-cylphospholipid (stabilized micelles) were utilized for flow cytometry tests to reveal the accumulation level of nanomicelles in the cells of various organs. The results showed that locally applied to the irradiation of the tumor with 1–3 MHz ultrasound during 30 s resulted in considerable accumulation of both types of Pluronic nanomicelles after intraperitoneal and intravenous injections. Thus, the selectivity of ultrasound-responsive Pluronic-based nanomicelles toward tumors cells was demonstrated.

The ultrasound-triggered and unaffected release of another hydrophobic dexamethasone drug from poly(2-oxazoline)-formed nanomicelles was investigated by Salgarella et al. [209]. Five different types of nanomicelles of the following content were utilized for studies: hydrophilic poly(2-methyl-2-oxazoline) and hydrophobic poly(2-n-propyl-2-oxazoline) or poly(2-butyl-2-oxazoline-co-2-(3-butenyl)-2-oxazoline). This study indicated the effect of the type of copolymer used and ultrasound wave on the amount of released drug. Along with stimulation time, such parameters resulted in a variation of dexamethasone release from 6% to 105%. In another study, peptide-based nanomicelles (PAIN) were loaded into Arg-Gly-Asp target compound in the presence of ICG as sono/photosensitizers to established SDT/PDT/PTT platform system (Figure 14A) [210]. The ROS generation and thermal impact of the RGD-PAIN activated by laser (808 nm) were evaluated (Figure 14B).

### 9.5. Redox Potential Responsive

The intracellular glutathione redox potential is considered to be 100-fold higher than its normal extracellular level. This makes glutathione an excellent target for drug delivery in cancer therapy [211,212,213,214,215]. The common strategy is to encapsulate drugs in nanomicelles formed by means of disulfide bonds. When approached to cancer cells, the elevated level of glutathione leads to the reduction of these bonds and thereby to drug release [216]. For example, the design of multi stimuli-responsive nanomicelles sensitive to redox potential, pH, and temperature was reported recently [217]. The hydrophobic part of the structure tetrahydropyran-protected 2-hydroxyethyl methacrylate aimed to respond acidity change, while the hydrophilic part provided by pNIPAM is temperature sensitive and with redox-potential sensitive disulfide bond between them. The conditions required for these nanomicelles to respond to stimuli are following: the temperature-sensitive hydrophilic part of nanomicelles is turned to hydrophobic above the low critical solution temperature that transfers the polymer to insoluble in water state which, in turn, leads to the assembly. The pH decrease, on the other hand, converts acid-sensitive hydrophobic part to hydrophilic and thereby inhibits the assembly. Finally, reducing disulfide bonds in such an environment results in the decomposition of nanomicelles [217].

Recently, Li’s group reported the design of amphiphilic block copolymer containing PEG which is prone to form nanomicelles [218]. Afterwards, DOX was encapsulated in these nanomicelles, where its release was induced by H_2_O_2_ and monitored via the change in the ratiometric fluorescence. Confocal laser scanning microscopy showed endocytosis of such fluorescent nanoparticles into A549 cells. As a result, DOX-loaded nanoparticles showed concentration-dependent cytotoxicity, whereas naked ones and their degradation residues were cytocompatible. Moreover, exogenous H_2_O_2_ and lipopolysaccharide-assisted stimulation enabled the remarkable diminishing of cancer cells.

A novel redox-responsive amphiphilic micellar system with a dual function of cancer chemotherapy and bioimaging was developed by Wang’s group [219]. The spherical nanomicelles were produced by self-assembly of amphiphilic copolymer tetraphenylethene (TPE) conjugated-ss-poly(aspartic acid)-block-poly(2-methacryloyloxyethyl phosphorylcholine) (TPE-SS-PLAsp-b-PMPC) and subsequently, DOX was encapsulated in them. Drug release and antitumor efficiency due to quick disassembly were studied after exposure to the high amounts of glutathione.

Alternative nanocarrier for drug delivery was fabricated with deoxycholic acid-grafted dextran (Dex-SSDCA) containing a disulfide bond [220]. The polymer was prone to self-assemble and form nanomicelles at concentrations less than 56 g/mL and could efficiently encapsulate DOX. Rapid disassembly was observed upon exposure to 10 mM Dithiothreitol (DTT) which led to the immediate release of DOX. In vitro studies showed that DOX-loaded Dex-SSDCA nanomicelles considerably diminished drug resistance of MCF-7/Adr cells and inhibited their growth. This drug delivery system demonstrated in vivo significant suppression of subcutaneous SKOV-3 ovarian cancer cells with notable inhibition of tumor angiogenesis and proliferation and enhanced apoptosis when compared with free DOX and negative control. Similarly, new nanomicelles were constructed via PEG-stearyl amine self-assembly [221]. To enhance the chemical and PDT treatment, Dox and photosensitive pheophorbide A are successfully encapsulated into the developed nanomicelles. Laser irradiation of the tumor cells results in fast dissociation of nanomicelles and rapid release of DOX (Figure 15).

Our group [21] prepared a novel biocompatible micelles-based phosphate for UCNPs targeting to prostate cancer cells. The developed micelles were found to be sPLA-2 enzyme responsive, as illustrated in Figure 16. The cytotoxicity results demonstrated that UCNPs-encapsulated phosphate micelles possess low toxicity toward prostate cancer cells.

## 10. Conclusions and Future Perspective

Developments in targeted drug delivery remain demanding. In this regard, designing various nanoscale carrier systems allows avoiding static and dynamic barriers in the human body. Among them, the application of nanomicelles for drug delivery purposes is trending upward in recent years because of their prominent and advantageous intrinsic properties. For instance, nanomicelles have high drug loading capacity and sustainable drug release, excellent water solubility and unique colloidal stability along with prominent low toxicity. Such properties allow us to use them to deliver therapeutics to anterior segment. They could also enable the application of nanomicelles in the prevention of drug resistance and reducing the toxicity of anticancer therapeutics. The possibility to vary the size of such nanocarriers by choosing surfactant or polymer with suitable length increases their EPR effect, which is crucially important in anticancer therapy. Despite enormous advances in the designing of nanomicelles, further studies in their application to the treatment are ongoing.

In the first part of this review, we have focused on the different class of polymeric materials that are used for preparing nanomicelles. Block copolymer components desired for producing amphiphilic block copolymers were described to help in the convenient selection of block copolymer segments for successful nanomicelles preparation. Although several polymeric materials were studied during the previous two decades, there are only a few polymeric micelles which have reached the clinical stage application. These include many hydrophobic polymers that are being used to develop the core of the polymeric micelles and just one hydrophilic polymer, PEG or PEO, used to prepare micelles shell, which is used in various nanomicelles in clinical investigations. There are some limitations to the use of the current materials including poorly soluble drug encapsulation in some cases, toxicity, and, in the case of PEG, unfavorable immunological interactions such as antibody responses. Therefore, the development of novel materials that are safe and enable high drug loading is highly needed. In our opinion POx- and POzi- based block copolymers satisfy these requirements and deserve future research, and more useful materials are likely to emerge in the future.

In the second part of this review, we extensively described the preparation of polymeric micelle for efficient delivery of poorly soluble anticancer drugs. While initially thought to solubilize based on simple hydrophobic interactions, recent advances in analytical methods have indicated new insights into these drug delivery systems. For intense, drug-polymer interactions are not simply limited to the hydrophobic blocks. In fact, hydrophilic, shell-forming blocks also play an important role in solubilization. Moreover, selective delivery is achieved via the design of various stimuli-responsive nanomicelles that release drugs based on endogenous or exogenous stimulations such as pH, temperature, ultrasound, light, redox potential, and others have been described also in this part. Thus, it is expected that further investigations of nanomicelles potential will contribute to major improvements in the efficiency of drug delivery systems. Selective and targeted delivery of anticancer drugs loaded into stimuli-responsive nanomicelles may eradicate the detrimental effect of conventional anticancer therapy on the human body.

It is important to point out important research areas and future directions that were intentionally left out of the current review’s concentrated consideration. The use of ionic block copolymers for drug delivery of natural biopolymers is one such area. A variety of research studies have contributed to the introduction of cationic block copolymers containing polycation blocks to bind negatively charged nucleic acids and water-soluble anionic blocks to ensure micelle stability in solution. Proteins, being polyampholytes, can also be formed with either cationic or anionic block copolymers into polymeric micelles. For the delivery of supramolecular biopolymer complexes, including oligomeric enzymes, multienzyme complexes or protein and nucleic acid complexes such as Cas9 and RNA guide, this technology can also be extended. In all these cases, the block copolymers’ polyelectrolyte blocks bind electrostatically with the oppositely charged molecules that form a polyion complex, which is usually insoluble and becomes segregated within the core of polymeric micelles. In aqueous environment, the hydrophilic blocks of these block copolymers form a shell around the core that stabilizes the micelles. In specific cases, hydrophobic interactions between the reacting molecules or the formation of hydrogen bonds between block copolymers and therapeutic molecules, in addition to electrostatic interactions, which play a key role in the self-assembly and stabilization of such polymeric micelles. These significant technologies have some key characteristics similar to the amphiphilic block copolymer micelles presented in this study but are also different in some fundamental aspects, such as formation mechanisms, stability, interactions with body fluid components, and cell entry, requiring a separate update and review.

In addition, the development of multifunctional nanomicelles bearing imaging, targeting, and stimuli-responsive agents might enable the increase in cancer therapy efficiency. However, additional studies should also be focused on revealing possible side effects of such approaches since the molecular and phenotypic heterogeneity in cancer cells may result in the lack of response to a single anticancer drug. This can be identified by diagnostic agents that are involved in theranostic nanomicelles–dual system with diagnostic and therapeutic moieties. Additionally, such versatile nanomicelles could be applied to personalized therapy of cancer and thus increase the efficiency of therapy.

## Figures and Tables

**Figure 1 nanomaterials-11-00070-f001:**
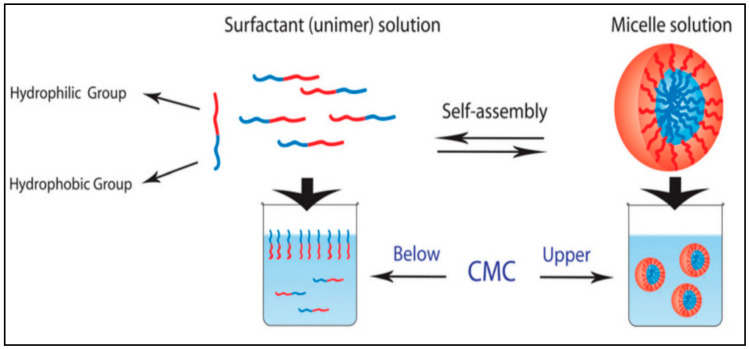
Schematic representation of unimer–micelle equilibrium in water. Reprinted with permission from [33]. Copyright © 2017, The Royal Society of Chemistry.

**Figure 2 nanomaterials-11-00070-f002:**
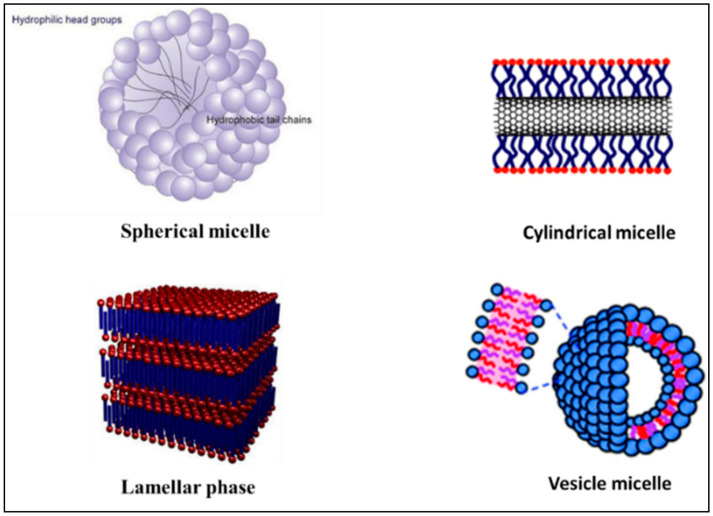
Illustration of some different self-assembled micelles structure. Reprinted with permission from [41]. Copyright © 2017, The Royal Society of Chemistry.

**Figure 3 nanomaterials-11-00070-f003:**
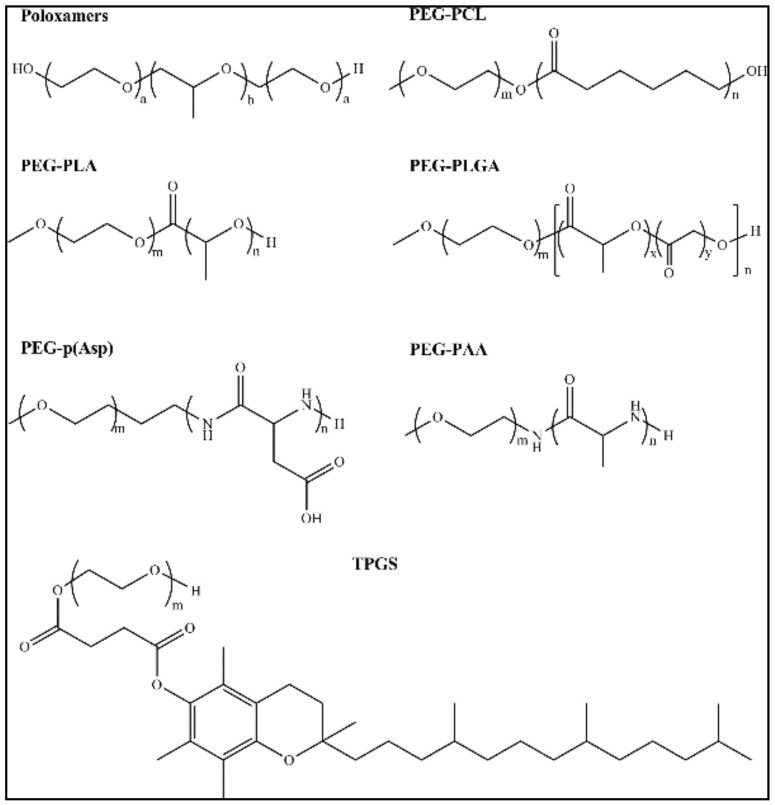
Structures of some commonly used polymers as micellar carrier: poly(ethylene glycol)-poly(ε-caprolactone) copolymers (PEG-PCL); poly(ethylene glycol)-b-poly(l-lactide) (PEG-PLA); Poly(ethylene glycol) methyl ether-block-poly(lactide-co-glycolide) (PEG-PLGA); Poly(ethylene glycol)-block-poly(l-aspartic acid) (PEG-p(Asp); poly(ethylene glycol) poly(amino acid) (PEG-PAA);, d-α-tocopheryl polyethylene glycol 1000 succinate (TPGS). Reprinted with permission from [52]. Copyright © 2018, American Chemical Society.

**Figure 4 nanomaterials-11-00070-f004:**
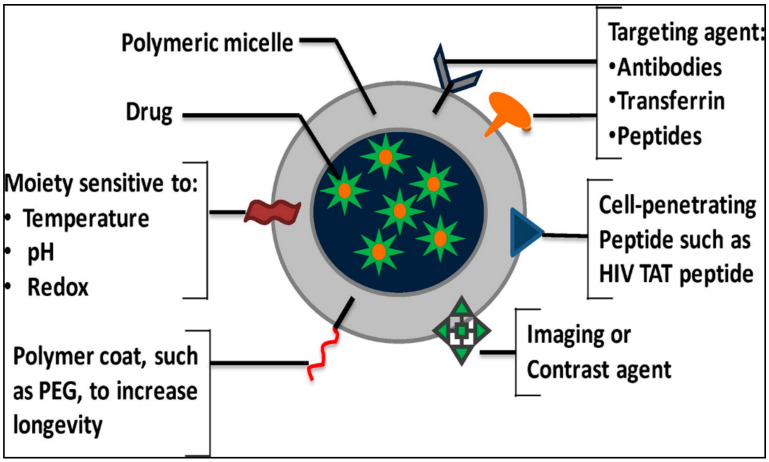
Schematic illustration of the polymeric nanomicelles targeting for drug delivery Reprinted with permission from [105].Copyright © 2016, Elsevier.

**Figure 5 nanomaterials-11-00070-f005:**
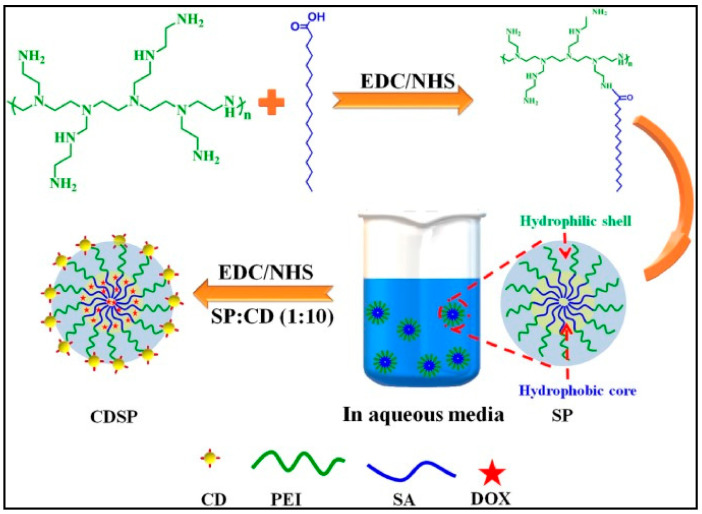
Schematic representation of carbon dots functionalized stearic-g-polyethyleneimine synthesis. Reprinted with permission from [127]. Copyright © 2021, Elsevier.

**Figure 7 nanomaterials-11-00070-f007:**
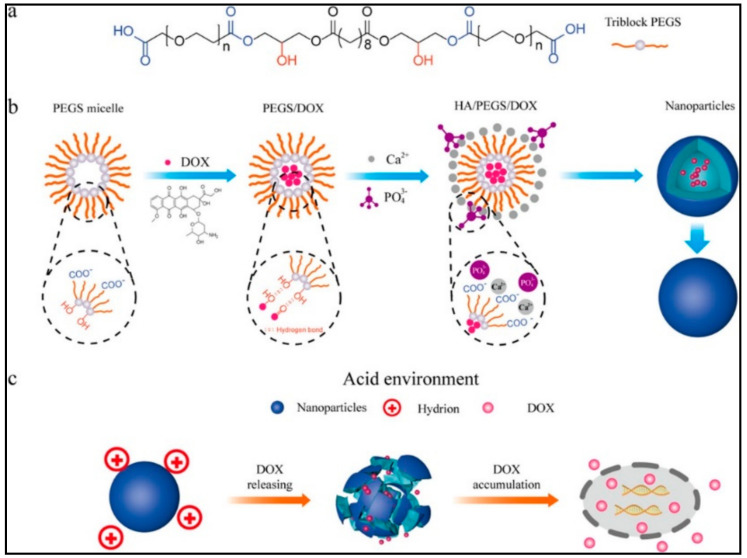
(**a**) Chemical Structure of the prepared polymer (PEGS). (**b**) Preparation strategy of PEGS/HA/DOX delivery system. (**c**) DOX release mechanism in acidic tumor environment. Reprinted with permission from [161]. Copyright © 2020, American Chemical Society.

**Figure 8 nanomaterials-11-00070-f008:**
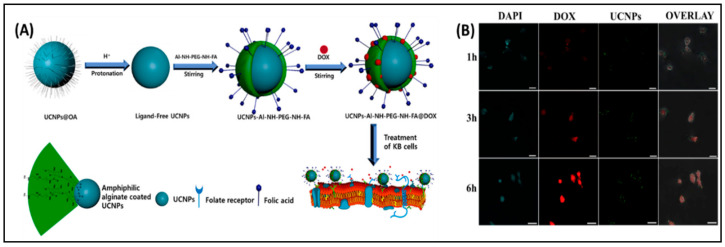
(**A**) Preparation of UCNP-Al-NH-PEG-NH-FA nanomicelles (**B**) Bioimaging results of KB cells treated with DOX-loaded UCNP-Al-NH-PEG-NH-FA for different time (1 h, 3 h, and 6 h) at 37 °C. Reprinted with permission from [17]. Copyright © 2018, Elsevier.

**Figure 10 nanomaterials-11-00070-f010:**
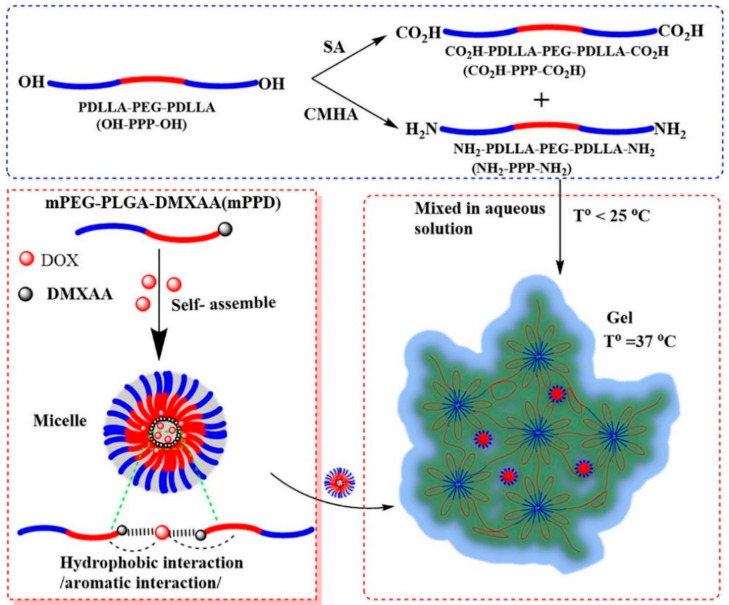
Schematic illustration of the thermoresponsive hydrogel-micelles for DOX delivery and cancer treatment. Reprinted with permission from [177]. Copyright © 2021, Elsevier.

**Figure 11 nanomaterials-11-00070-f011:**
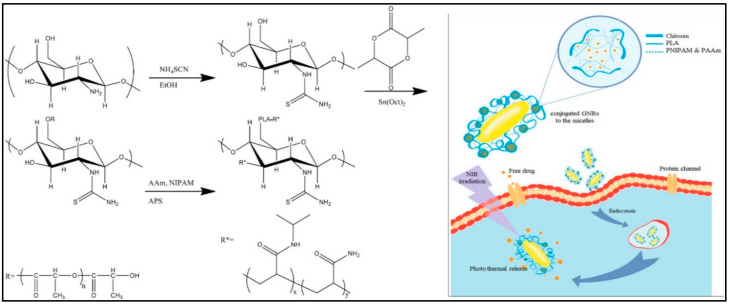
Synthesis process of polymeric micelles of CS-S-PLA-PNIPAM-*co*-PAAm and light triggered DOX release and imaging for cancer cells using copolymer micelles conjugated Au-NIR probe. Reprinted with permission from [187]. Copyright © 2020, Elsevier.

**Figure 12 nanomaterials-11-00070-f012:**
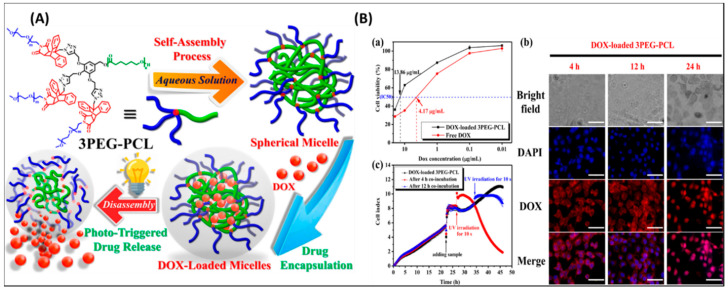
(**A**) The design strategy of 3PEG–PCL micelles for DOX loading and release strategy. (**B**) (**a**) toxicity test of human oral squamous cell carcinoma (SAS) cells. (**b**) Bioimaging results of SAS cells treated with DOX and DOX-encapsulated micelles. (**c**) Cytotoxicity profiles calculated form real time analyzer. Reprinted with permission from [189]. Copyright © 2019, American Chemical Society.

**Figure 14 nanomaterials-11-00070-f014:**
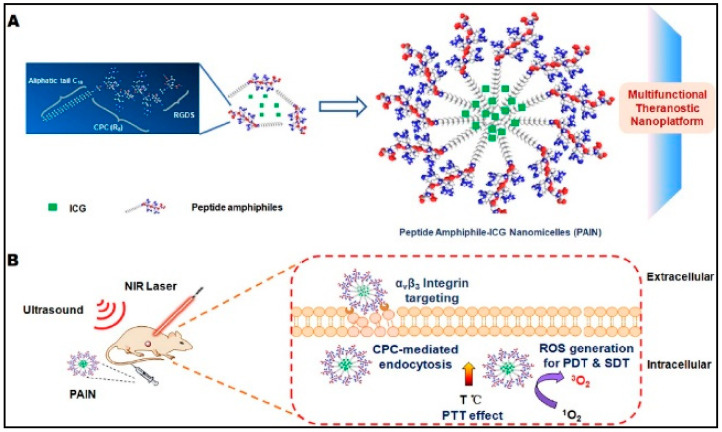
(**A**) Self-assembled of the prepared peptide-based ICG nanomicelles and (**B**) the NIR and ultrasound irradiation effects on the tumor mice. Reprinted with permission from [210]. Copyright © 2020, Elsevier.

**Figure 15 nanomaterials-11-00070-f015:**
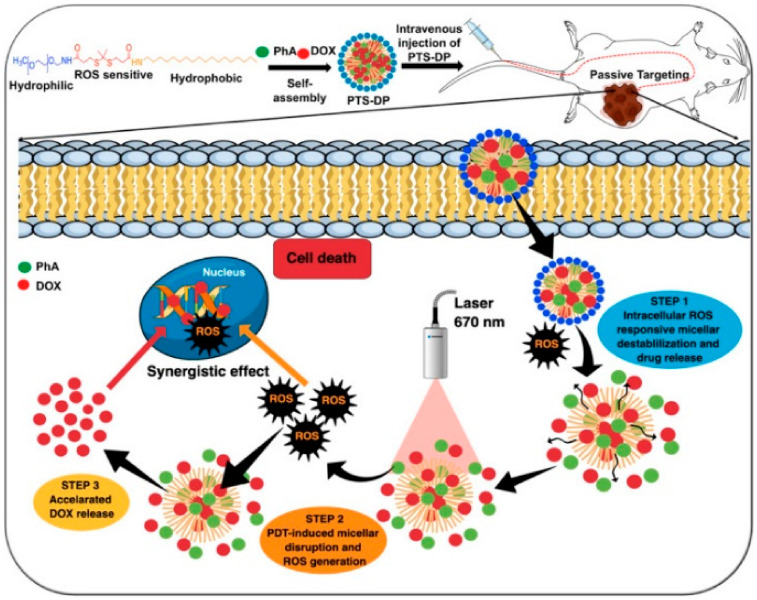
Preparation of PTS-DP nanomicelles and their DOX loaded and release strategy under ROS trigger for enhanced chemo-photodynamic therapy. Reprinted with permission from [221]. Copyright © 2020, Elsevier.

**Figure 16 nanomaterials-11-00070-f016:**
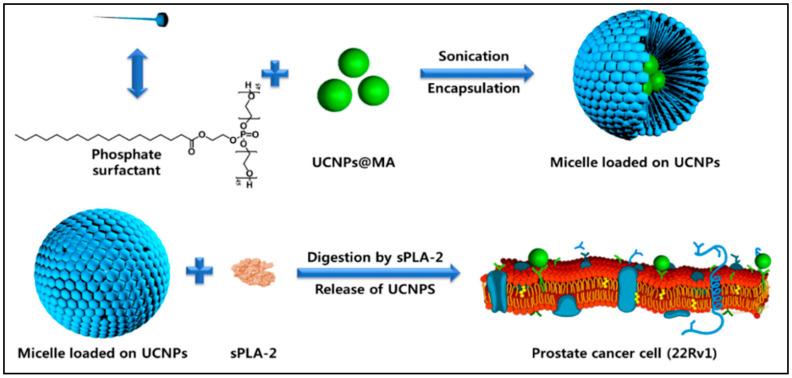
Preparation of phosphate micelle coated UCNPs and the UCNPs release mechanism under sPLA-2 enzyme trigger. Reprinted with permission from [21]. Copyright © 2017, Springer Nature.

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
