# Peer review of "Recent Advances in Nanomicelles Delivery Systems"

_nanomaterials, 2020, doi:10.3390/nano11010070_

Round 1
Reviewer 1 Report
Tawfik et al. present a well structured review on the use of nanomicelles for drug delivery. The reader is guided easily through the text and a substantial amount of figures is presented. A wide range of examples is discussed based upon the stimulus that leads to the release of the included drug. Overall, the reader gets a good impression of what is currently possible in the field.
The work could be further improved by enhancing the introductory parts on nanomicelles and polymer nanomicelles with more detailed descriptions and discussions of their synthesis so that the reader could more easily decide which type of polymer or micelle could be useful for their delivery interests. This should include more details on the polymers, the influence of their molecular weight, influences of solvents on micelle sizes, description of the synthetic methods and their differences, as well as details on their metabolism / biodegradation in biological tissues and in vivo.
Equally the conclusion of the review could go a bit more into detail comparing advantages and disadvantages of the presented delivery systems and release stimuli. This would be of great benefit to the reader.
Generally, a lot of the discussed examples could be presented more in depth so that the advantage upon delivery by micelles can be better appreciated.
For a detailed look at minor points please look at the attached pdf file. I recommend acceptance after minor revisions.

Author Response
We greatly appreciate the reviewers for going through the manuscript and making such insightful comments and suggestions that have undoubtedly helped in improving the quality of the manuscript. Please find below our responses to the comments and suggestions of reviewers. The manuscript was revised carefully (with highlighted red in the revised manuscript). We hope this version could meet the standards of the referees and the journal.
Reviewer #1:
Tawfik et al. present a well structured review on the use of nanomicelles for drug delivery. The reader is guided easily through the text and a substantial amount of figures is presented. A wide range of examples is discussed based upon the stimulus that leads to the release of the included drug. Overall, the reader gets a good impression of what is currently possible in the field.
Response: Many thanks for these constructive comments and useful suggestions. According to the suggestions, we carefully revised the manuscript. Point-by-point responses to the comments and suggestions are listed below.
Comment #1: The work could be further improved by enhancing the introductory parts on nanomicelles and polymer nanomicelles with more detailed descriptions and discussions of their synthesis so that the reader could more easily decide which type of polymer or micelle could be useful for their delivery interests. This should include more details on the polymers, the influence of their molecular weight, influences of solvents on micelle sizes, description of the synthetic methods and their differences, as well as details on their metabolism / biodegradation in biological tissues and in vivo.
Response: Very thanks for your comment. Polymeric micelles, polymer selection, synthesis, and various properties of polymers was described in details and more discussion were added to the text. Please see pages 5-10.
Comment #2: Equally the conclusion of the review could go a bit more into detail comparing advantages and disadvantages of the presented delivery systems and release stimuli. This would be of great benefit to the reader.
Response: Very thanks for your comment and advice. In the conclusions section more description for the advantages of the presented review was added in details. Please see page 27- 28.
Comment #3: Generally, a lot of the discussed examples could be presented more in depth so that the advantage upon delivery by micelles can be better appreciated.
Response: Very thanks for your comment and advice. More discussion and examples about the advantage of using micelles-based delivery systems have been added in the revised review.
Comment #4: For a detailed look at minor points please look at the attached pdf file. I recommend acceptance after minor revisions.
Response: Very thanks for your positive comment. The manuscript was carefully revised based on the correction points recommended in the attached pdf file.
Reviewer 2 Report
In this manuscript, the authors presented a review of recent developments on naonomicelles delivery systems in therapeutic applications. The review had a discussion on the nanomicelles in terms of categories, properties, preparation, targeting properties, and various stimuli-responsive characteristics. This review summarized the recent kinds of literature on nanomicelles drug delivery systems and conveyed the information on the significance of precise therapy and efficient treatment in tumors/cancers. Nevertheless, there is still a long way to go.
There are some issues that need to be addressed for minor revision:
- In line 115, authors mentioned ‘the longer circulations in bloodstream’, should be clearly demonstrated how long the nanomicelles can circulate in bloodstream and compared with what control group can say it has better circulation property in terms of time?
- In part 6. nanomicelles delivering anticancer drugs, authors only discussed on DOX samples, can authors list any other anti-tumor drugs used in nanomicelles system except DOX?
- In part 8.1 pH-sensitive, can authors summarize the pH-sensitive linkers used in nanomicelles delivery systems in most recent literatures?
- In part 8.3 light-sensitive, NIR irradiation is better than UV in terms of minimized harm to tissues and deep penetration depth, can authors present and list some examples in form of figures of in vivo imaging in animal model with deep tumor?
- Elaboration on the importance of this study is not enough, opinions about prospects and development of the nanomicelles delivery systems from authors are not very insightful. Reorganizing the conclusions and future perspectives part is highly recommended.
- Some format issues should be revised: Line 26 should be aligned text to the left; Line 149,397,447,500,504,576 should be text-indented. In addition, ‘bar’ in line 327 should be ‘bare’, other word-spelling mistakes need to be double-checked.
- Several figures should be rearranged, e.g., Fig. 8 is compressed, the word in Fig. 13(B) a&b is hard to read.
In general, I would like to recommend this review for publication in Nanomaterials after minor revision.
Author Response
We greatly appreciate the reviewers for going through the manuscript and making such insightful comments and suggestions that have undoubtedly helped in improving the quality of the manuscript. Please find below our responses to the comments and suggestions of reviewers. The manuscript was revised carefully (with highlighted red in the revised manuscript). We hope this version could meet the standards of the referees and the journal.
Reviewer #2:
In this manuscript, the authors presented a review of recent developments on naonomicelles delivery systems in therapeutic applications. The review had a discussion on the nanomicelles in terms of categories, properties, preparation, targeting properties, and various stimuli-responsive characteristics. This review summarized the recent kinds of literature on nanomicelles drug delivery systems and conveyed the information on the significance of precise therapy and efficient treatment in tumors/cancers. Nevertheless, there is still a long way to go.
There are some issues that need to be addressed for minor revision:
Response: Thanks for your positive comments and constructive suggestions. We have carefully revised the manuscript. Point-by-point responses to the comments and suggestions are listed below.
Comment #1: In line 115, authors mentioned ‘the longer circulations in bloodstream’, should be clearly demonstrated how long the nanomicelles can circulate in bloodstream and compared with what control group can say it has better circulation property in terms of time?
Response: very thanks for your comment. More discussion about circulations in bloodstream using the nanomicelles and the circulation property in terms of time was added in the revised review. Please see page 3-4.
Comment #2: In part 6. nanomicelles delivering anticancer drugs, authors only discussed on DOX samples, can authors list any other anti-tumor drugs used in nanomicelles system except DOX?
Response: very thanks for your comment. Some other anti-tumor drugs used in nanomicelles system except DOX were listed and discussed in part 6. nanomicelles delivering anticancer drugs. Please see page 11.
Comment #3: In part 8.1 pH-sensitive, can authors summarize the pH-sensitive linkers used in nanomicelles delivery systems in most recent literatures?
Response: very thanks for your comment. Several drug release mechanisms, pH-sensitive linkers-based drug delivery system was summarized in part 8.1 as recommended. Please see pages 13-15.
Comment #4: In part 8.3 light-sensitive, NIR irradiation is better than UV in terms of minimized harm to tissues and deep penetration depth, can authors present and list some examples in form of figures of in vivo imaging in animal model with deep tumor?
Response: very thanks for your comment. Some examples for in vivo imaging in animal model and NIR penetration were presented in pages 21-23. Please see figure 13.
Comment #5: Elaboration on the importance of this study is not enough, opinions about prospects and development of the nanomicelles delivery systems from authors are not very insightful. Reorganizing the conclusions and future perspectives part is highly recommended.
Response: very thanks for your comment and advice. More details and discussion about the development of the nanomicelles delivery systems were presented in the conclusions section.
Comment #6:. Some format issues should be revised: Line 26 should be aligned text to the left; Line 149,397,447,500,504,576 should be text-indented. In addition, ‘bar’ in line 327 should be ‘bare’, other word-spelling mistakes need to be double-checked.
Response: Thanks for your constructive suggestions. We have carefully revised the manuscript. Point-by-point and all mistakes were corrected.
Comment #7: Several figures should be rearranged, e.g., Fig. 8 is compressed, the word in Fig. 13(B) a&b is hard to read.
Response: Thanks for your comment. All figures were rearranged and corrected.
Comment #8:In general, I would like to recommend this review for publication in Nanomaterials after minor revision.
Response: Very thanks for your positive comment.